# Neuronal Intranuclear Inclusion Disease-Related Neurotrophic Keratitis: A Case Report

**DOI:** 10.3390/brainsci12060782

**Published:** 2022-06-14

**Authors:** Pei Liu, Xuemei Lin, Xiangjun Chen, Tor Paaske Utheim, Wei Gao, Yan Yan, Songdi Wu

**Affiliations:** 1Department of Neuro-Ophthalmology, The First Affiliated Hospital of Northwest University, Xi’an 710082, China; liupei32@icloud.com (P.L.); xuemei_yilei@163.com (X.L.); 2Department of Ophthalmology, Sørlandet Hospital Arendal, 4838 Arendal, Norway; chenxiangjun1101@gmail.com (X.C.); utheim2@gmail.com (T.P.U.); 3Department of Ophthalmology, Vestre Viken Hospital Trust, 3004 Drammen, Norway; 4Department of Medical Biochemistry, Oslo University Hospital, 0450 Oslo, Norway; 5Department of Ophthalmology, Xi’an People’s Hospital, Xi’an 710100, China; gaoweiz@163.com; 6Department of Pathology, The First Affiliated Hospital of Northwest University, Xi’an 710082, China; yanyan03281@163.com

**Keywords:** neuronal intranuclear inclusion disease, ocular environment, neurotrophic keratitis, in vivo confocal microscopy, diffusion-weighted imaging

## Abstract

Neuronal intranuclear inclusion disease (NIID) is a rare and slowly progressive neurodegenerative disease characterized by the presence of eosinophilic neuronal intranuclear inclusions. The clinical manifestations of NIID are diverse, and the most common initial feature in cases of sporadic NIID is dementia. Herein, we report an adult female with keratitis as the initial presentation with subsequent bilateral limb tremor, gait disturbances, overemotional behavior, sweating and constipation. Diffusion-weighted imaging (DWI) showed hyperintensity in the bilateral fronto-parieto-occipital corticomedullary junction. Skin biopsy specimens revealed eosinophilic hyaline intranuclear inclusions in fibroblast cells, sweat gland cells and adipose cells. In vivo confocal microscopy of the cornea indicated the absence of corneal nerves in both affected eyes. The patient’s diagnosis of NIID was based on the presence of intranuclear inclusions in biopsied skin and the characteristic high-intensity signal in the corticomedullary junction obtained with DWI. This case report emphasizes that the clinical heterogeneity of NIID and an examination of the corneal nerves may offer valuable clues to its early diagnosis in some patients.

## 1. Introduction

Neuronal intranuclear inclusion disease (NIID) is a rare and slowly progressive neurodegenerative disease characterized by the presence of eosinophilic and ubiquitin-positive neuronal intranuclear inclusions in the central and peripheral nervous systems, as well as in visceral organs [1,2]. NIID is divided into infantile, juvenile and adult forms based on the age of onset, with the adult form being the most common subtype in East Asia [2]. An abnormal expansion of GGC repeats in the NOTCH2NLC gene was recently identified as the possible cause of sporadic NIID [3,4,5]. The clinical manifestations of NIID are various, involving progressive cognitive impairment, epilepsy, encephalopathy, limb weakness, Parkinson syndrome, peripheral neuropathy and autonomic dysfunction. Dementia has been reported to be the initial and primary clinical manifestation in most sporadic NIID cases [6].

The ocular manifestations of NIID have recently attracted attention and have expanded the clinical spectrum of NIID in neuro-ophthalmology. The novel ophthalmological manifestations of the disease involve miosis, night blindness, oculogyric crisis, reduced eye movements, nystagmus, blepharospasm, ptosis and rod–cone dysfunction with progressive outer retinal degeneration [7]. In a previous study, corneal nerve fibers were quantified using in vivo confocal microscopy (IVCM) in a small sample of patients with adult-onset NIID without ocular complaint [8]. The large nerve fibers were normal, whereas the small fibers were loose and tortuous, suggesting that corneal changes may precede neurological manifestations [8].

In the present article, we report for the first time a case of sporadic adult-onset NIID with neurotrophic keratitis as the initial manifestation and fewer corneal nerve fibers, as detected with IVCM. The purpose of this paper is two-fold: (1) to raise awareness that neurotrophic keratitis is an initial manifestation of NIID and (2) to describe the corneal morphological presentations that can contribute to the diagnosis of NIID-related ophthalmopathy.

## 2. Case Report

In April of 2016, a 57-year-old female was admitted with progressively decreasing vision in the right eye over the previous 8 years. Eye pain and redness, photophobia and lacrimation were also reported. The patient was diagnosed with viral keratitis at a local hospital on admission. Despite receiving antiviral therapy and an amniotic membrane transplant in the right eye, the patient’s vision continued to worsen. The patient denied a history of diabetes mellitus, trauma and stroke. She also did not report any other ocular manifestations, such as rod–cone dysfunction, retinal degeneration, night blindness and miosis, during the 8 years prior to being admitted to the hospital.

Upon examination, the best-corrected visual acuity was light perception in the right eye and 10/20 based on the Snellen chart in the left eye. Slit-lamp microscopy revealed conjunctival hyperemia, nubecula, keratoleukoma and neovascularization of the cornea in the right eye (Figure 1A). A cotton-tipped applicator was used to reveal reduced corneal sensitivity of the right eye. The size and shape of the pupil and the fundus of the right eye could not be examined due to opacities on the cornea. Slit-lamp microscopy of the left eye showed nasal conjunctiva hyperemia and pseudopterygium extending onto the cornea; otherwise, the cornea and pupil were normal (Figure 1B). IVCM of the right cornea using a confocal laser scanning microscope (HRT II, Rostock Corneal Module, Heidelberg Engineering, Heidelberg, Germany) revealed the presence of intraepithelial Langerhans cells and the absence of the subbasal corneal nerve plexus (Figure 1E). Stromal edema with the infiltration of inflammatory cells and a stromal scar were also detected (Figure 1F–H). The patient underwent lamellar corneal transplantation in the right eye in May 2016 because of the protracted course of keratopathy and leukoma. Thereafter, the patient underwent repeated lamellar keratoplasty in July of 2006 because of transplant rejection. 

In May of 2018, the patient was re-admitted to the ophthalmic inpatient department because she noticed reduced visual acuity in the left eye over the previous 4 months. The sensitivity of the left cornea was reduced, as demonstrated with corneal reflex testing. Slit-lamp microscopy revealed a distinct corneal scar in the right corneal transplant with nubecula and vascularization (Figure 1C). Additionally, macula of the cornea with conjunctival hyperemia was observed in the epithelium of the left eye (Figure 1D). IVCM of the left cornea showed edematous and disorganized epithelial cells (Figure 1I,J), and the nerve fibers were almost absent; Langerhans cells were found in the corneal basal epithelial layer (Figure 1K), whereas the endothelial cells generally appeared normal (Figure 1L). Combining the information of a protracted course of keratitis, IVCM alterations and normal endothelial density and morphology, the patient was diagnosed as having left neurotrophic keratitis. Due to the overwhelming corneal damage threatening the left eye, an amniotic membrane transplantation was performed. A viral antigen test and bacterial and fungal cultures were negative. Histopathological analysis of the right cornea indicated chronic nonspecific inflammation.

During her second hospitalization, the patient consulted with a neurologist because of some neurological symptoms, including bilateral limb tremor, gait disturbances, overemotional behavior, sweating and constipation, which may have existed for the previous 2 years. There was no bradykinesia, rigidity, limb weakness or numbness. The patient’s body mass index was 22.9 kg/m^2^. The patient was oriented and had fluent speech. Her cognitive function was intact. The left eye had a normal pupillary reaction to light and fundoscopy, whereas the right pupil could not be visualized because of decreased corneal transparency. The patient’s strength and sensation of the trunk and the limbs were normal. The patient had mild hypermyotonia and a resting tremor in the jaw, hands and feet. The patient had dysdiadochokinesia and deficits in the bilateral finger-to-nose test (indicating bilateral dysmetria) and the heel–knee–tibia slide test. The Romberg test was positive, but there was no Babinski or Chaddock sign. The remainder of the examinations were unremarkable.

Similar to the assessment of neurological signs and symptoms, neuropsychiatric and autonomic functions were also evaluated. The Hamilton Anxiety Scale score was 15 (definite anxiety), and the Hamilton Depression Scale score was 26 (definite depression). The Mini-Mental State Examination score was 30 (normal > 30), and the Montreal Cognitive Assessment score was 26 (normal ≥ 26). The Neuropsychiatric Inventory (NPI) score was 24 (no neuropsychological disorder). Autonomic (SCOPA-AUT) score was 11 (sometimes dysphagia, sometimes constipation, often sweating and often heat sensitive). There were no obvious abnormalities in the laboratory tests, including complete blood count, urine and stool analyses, liver and kidney function, and lipid profile, and blood glucose concentration was unremarkable. The following indicators were negative: infectious indicators, including HIV, treponema pallidum and hepatitis virus; purified protein derivative test; galactomannan test; glucan test; thyroid hormone test; immunological index, including anti-nuclear antibody (ANA), anti-neutrophil cytoplasmic antibody (ANCA), immunoglobulin (Ig) A, IgG, IgM, IgE, complement-3/4, and λ and κ light chain immunoglobulins; and tumor markers, including carbohydrate antigen 724 (CA724), carcinoembryonic antigen (CEA), alphafetoprotein (AFP), neuron-specific enolase (NSE), CA19-9, CA125, CA153, β-human chorionic gonadotrophin (hCG), cytokeratin-19-fragment (CYFRA21-1), CA50 and human epididymis protein 4 (HE4); the risk of ovarian malignancy algorithm (ROMA); and squamous cell carcinoma antigen (SCCAg). The axial planes of T2-weighted images of the brain taken using magnetic resonance imaging (MRI) revealed symmetrically diffuse and high signal leukodystrophy. Such areas presented isointense or slight hypointensities in T1-weighted imaging. The diffusion-weighted imaging (DWI) sequence showed a high-signal-intensity zigzag edging sign in the bilateral fronto-parieto-occipital corticomedullary junction, also known as the “cortical line sign” (Figure 2). The corresponding lesions showed isointense or slightly hyperintense signals on the ADC sequence. The CGG repeat number in the fragile X chromosome mental retardation gene 1 (FMR1) gene was normal. Therefore, fragile X-associated tremor/ataxia syndrome could be excluded. Skin biopsy specimens revealed eosinophilic hyaline intranuclear inclusions (Figure 1M–O) that were strongly positive for anti-ubiquitin (Figure 1N) and anti-p62 antibodies (Figure 1O) in fibroblast cells, sweat gland cells and adipose cells. The pathological staining of the previous corneal sections was also performed, but intranuclear inclusions were not observed in the epithelial or stromal nucleus of the cornea.

The typical “cortical line sign” in DWI of the brain combined with eosinophilic hyaline intranuclear inclusions in the skin biopsy supported the diagnosis of NIID. No family member had similar symptoms or the FMR1 gene premutation. The patient was finally diagnosed with sporadic NIID. With regard to the corneal impairment, we reviewed the pathogenic factors of the secondary neurotrophic keratitis along the pathways of the corneal sensory nerve, including the following causes: (1) injuries of corneal sensory nerves induced by viral infection, refractive surgery, chemical or thermal burns, ocular medication and wearing contact lenses; (2) damage of the ophthalmic branch of the trigeminal nerve, as well as ciliary nerves, due to tumor formation and surgeries (such as cataract and retinal surgeries or ophthalmic laser procedures); (3) trigeminal ganglion lesions due to herpetic viral infections, surgery for trigeminal neuralgia, preganglionic trigeminal nerve root injury (including acoustic neurinoma, schwannoma and aneurysms) and some systemic diseases (such as diabetes and leprosy) that can decrease sensory nerve function or damage sensory fibers; and (4) damage to the lowest part of the spinal trigeminal nuclei due to dorsolateral medullary infarction and tumor formation [9,10]. However, our patient developed keratopathy without any of the abovementioned histories, and neurotrophic keratitis due to the autonomic nervous dysfunction of NIID was established. The patient had a regular follow-up visit, but she was distressed because the neurological symptoms progressed, and the decreased vision affected every aspect of her life.

## 3. Discussion

NIID exhibits highly variable clinical manifestations, and the initial manifestation in adult-onset NIID is usually cognitive impairment, often accompanied by pyramidal signs and peripheral neuropathy with or without autonomic nervous system dysfunction [1,2]. The diagnosis of NIID is often delayed because of its insidious onset and nonspecific initial symptoms [2]. In this study, we focused on neurotrophic keratitis as the initial presentation in a patient with NIID. To the best of our knowledge, this is the first report of a 2-year longitudinal evaluation using IVCM of the neurotrophic keratopathy seen in NIID.

Previous studies reported several ophthalmological manifestations in cases of familial NIID, including miosis, night blindness, oculogyric crisis, eye movement disorders, nystagmus, blepharospasm, ptosis and retinopathy [1,7,11,12]. NIID presents with or without vision impairment, and the abnormal electroretinogram (ERG) in NIID is associated with rod–cone dysfunction and progressive retinal degeneration in the peripapillary and midperipheral regions [12,13]. 

To date, there is no evidence for the onset of NIID being marked with impaired vision caused by keratitis. In the current case, the intractable corneal epithelial defect was initially thought to be caused by a corneal infection. However, the affected cornea was negative for the presence of microbes, and IVCM revealed its structural abnormality but not signs of infection. Furthermore, the corneal exfoliation and opacification in the epithelium and the reduction of corneal nerves were not consistent with corneal degeneration and congenital dystrophy but were consistent with corneal neurotrophic dystrophy. The deteriorating keratopathy evolved during the progression of the disease, and the eyes were threatened despite the fact that the patient received several amniotic membrane transplants and underwent keratoplasties. We therefore attributed the lingering keratitis to neurotrophic keratitis of autonomic nervous dysfunction of NIID. In fact, autonomic innervation (mostly sympathetic nerves and a paucity of parasympathetic nerves) of the cornea played a vital role in maintaining the quality and quantity of tears and established homeostasis of the ocular environment [10,14,15]. The disruption of sympathetic corneal innervation thus contributes to neurotrophic keratitis [10,14,15]. Using corneal IVCM, Liu et al. [8] revealed fewer corneal nerves in the eyes of asymptomatic patients with NIID, an effect that worsened as the disease advanced. They suggested that the NIID-related corneal nerve changes detected by IVCM might precede the systemic neurological manifestations and, thus, may assist early treatment and the staging of the disease [8]. 

Biopsy is recommended as an effective and less invasive tool to diagnose NIID [1,16,17]. Ultrastructural neuronal intranuclear inclusions (NIIs) consisting of filamentous material without a restrictive membrane mixed with granular substances are found in the central and peripheral nervous systems, visceral organs, skin, rectal cells, renal tubule epithelium, adrenal medulla, hepatocytes, pancreatic cells, cardiomyocytes, fibroblasts and smooth and skeletal muscle cells [16,18]. The NIIs in NIID have immunohistochemical properties similar to those seen in poly Q diseases [19], which are a group of neurodegenerative diseases caused by polymorphic fragment elongation of the CGG triplet in FMR1 [20,21]. Similar to poly Q diseases, NIID is characterized by degeneration, but it has a different cause: NIIs are thought to be due to the accumulation of misfolded proteins that form insoluble aggregates or the dysfunction of ubiquitin-mediated protein degradation in the nuclear body. These processes may chronically induce cytotoxicity and cause neurodegeneration [22]. In the early stage of the disease, the factor that threatens the patient’s vision might be the persistent autonomic impairment of NIID to the corneal nerves. The lingering keratitis and repeated corneal operations led us to perform a biopsy of the cornea, but eosinophilic hyaline NIIs were not observed in the affected cornea. This finding may indicate that the corneal impairment was due to the dysfunction of autonomic nerves, rather than the cytotoxicity of NIIs, as the cause of corneal neurodegeneration. Furthermore, IVCM revealed the loss of corneal nerves in the involved eyes. More cases should be examined to elucidate the pathogenesis of NIID-related corneal impairment.

In brain imaging studies, a high-signal-intensity zigzag edging sign in the bilateral corticomedullary junction on a DWI sequence is considered as the hallmark of NIID. It reveals a neuropathological result of spongiotic changes proximal to the U-fibers or diffuse myelin pallor in subcortical white matter. This type of leukoencephalopathy may always be an anticipatory clue for clinicians to explore whether causative intranuclear inclusions are seen upon biopsy [23].

## 4. Conclusions

We herein report a case of sporadic adult-onset NIID with autonomic impairment manifesting as neurotrophic keratitis. Neurotrophic keratitis may serve as the initial presentation and the most prominent ophthalmological characteristic in NIID. The absence of NIIs in the affected cornea may indicate that the corneal impairment is due to the dysfunction of autonomic nerves rather than the cytotoxicity of NIIs in corneal neurodegeneration. This paper strengthens the awareness that quantitative NIID-related changes in the corneal nerves detected using IVCM may precede the systemic neurological manifestations and, thus, may assist in the early treatment and staging of the disease. Future studies are warranted to reveal the underlying mechanism of keratitis in patients with NIID.

## Figures and Tables

**Figure 1 brainsci-12-00782-f001:**
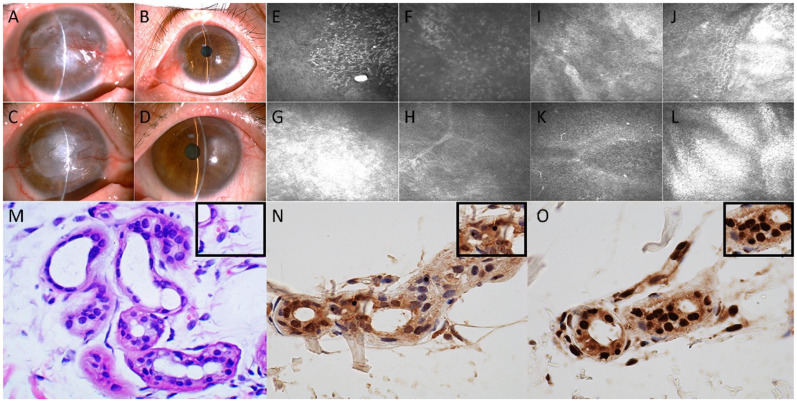
Anterior segment photography obtained in May of 2016 (**A**,**B**) and May of 2018 (**C**,**D**). In vivo confocal microscopy (IVCM) of the right eye demonstrated alterations in the epithelial basal cell layer of cornea (**E**) and stroma (**F**–**H**) in May of 2016, and alterations in the corneal epithelium (**I**,**J**), basal epithelial layer (**K**) and endothelium (**L**) in May of 2018. Hematoxylin–eosin staining (**E**,**H**,**M**) and immunohistochemical staining with anti-p62 antibody (**N**) and anti-ubiquitin antibody (**O**) of the skin biopsy with the magnified area showed the neuronal intranuclear inclusions.

**Figure 2 brainsci-12-00782-f002:**
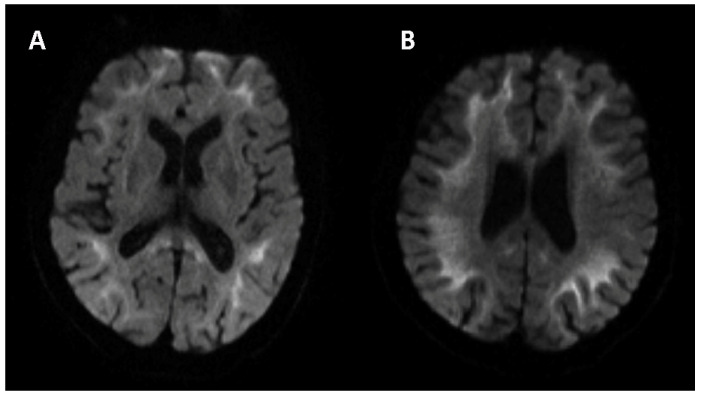
Brain diffusion-weighted imaging (DWI) sequence of magnetic resonance imaging (MRI) in the axial view showed high-signal-intensity zigzag edging sign in the corticomedullary junction which were exampled by the sections of basal ganglia (**A**) and centrum semiovale (**B**).

## Data Availability

Data are available upon request.

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
