# Peer review of "Neuronal Intranuclear Inclusion Disease-Related Neurotrophic Keratitis: A Case Report"

_brainsci, 2022, doi:10.3390/brainsci12060782_

Round 1

Reviewer 1 Report

Dear author,

the report was well presented therefore I suggested minor revision with a minor concern. However, in addition to my previous comment, there may other minor points to clarify:

1)Lumbar puncture should be performed in the setting of these neurologic manifestations (eg. neurofilament light chain levels, alfa synuclein etc). CNS infection and inflammatory diseases should be excluded. Please explain.

2) page 3 ln 110-111: “Her cognitive function was intact except for some short-term memory deficits”. How short-term memory was assessed? (eg. Tower of London). If there are no objective evidence of short-term memory deficits , this sentence should be removed. Short-term memory does not refers to a slight forgetfulness of recent events.

3) page 3 ln 125-127: “There were 125 no obvious abnormalities in the following: routine blood tests, biochemical examination 126 of blood plasma, infectious indicators, tumour markers, thyroid hormone test or immunological index.” You should report what exams were performed (eg. immunological index ? what index? ANA, ANCA?)

4) MRI: other sequences may be reported such as FLAIR, ADC, T2, T1 gd+. Indeed, DWI indicates the presence of recent ischemic stroke (cytotoxic edema) or demyelinating active lesions. Reporting the other MRI sequences would help the readership to better understand the nature of the disease.

4) page 5 ln 172-183: “We reviewed the pathogenic factors of the secondary neu- 173 rotrophic along the pathways of keratitis corneal sensory nerve, including the following 174 causes: (1) injuries of corneal sensory nerves induced by viral infection, refractive surgery, chemical or thermal burns, ocular medication and wearing contact lenses; (2) damage of 176 the ophthalmic branch of the trigeminal nerve as well as ciliary nerves due to tumour 177 formation and surgeries (such as cataract and retinal surgeries or ophthalmic laser procedures); (3) trigeminal ganglion lesions due to herpetic viral infections, surgery for trigeminal neuralgia, preganglionic trigeminal nerve root injury (including acoustic neurinoma, 180 schwannoma and aneurysms) and some systemic diseases (such as diabetes and leprosy) 181 that can decrease sensory nerve function or damage sensory fibres and) damage to the lowest part of spinal trigeminal nuclei due dorsolateral medullary infarction and tumour formation”. The diagnostic process should be reported in the section of “2.Case Report (eg. results of viral serology, HSV, VZV) thus this paragraph may be shortened.

5) page5 ln 220-221 “For example, the premutation of the FMR1 gene leads to neuro- 220 degeneration and the development of intranuclear inclusions in NIID [22, 24]. “ Ref n 24 is lacking in the manuscript. However, this sentence may be deleted since you have already said it elsewhere in the manuscript.

6) language revision /grammar check

If GGC repeats of the NOTCH2NLC were found to be associated with sporadic, Why did not you perform genetic analysis ? 

Thank you  

Author Response

We gratefully acknowledge the comments raised by the the reviewer. These remarks have helped to improve the manuscript significantly. The comments have been carefully considered and addressed point-by-point below.

Point 1: Lumbar puncture should be performed in the setting of these neurologic manifestations (eg. neurofilament light chain levels, alfa synuclein etc). CNS infection and inflammatory diseases should be excluded. Please explain. 

Response 1: We thank the reviewer for the constructive comment. We agree with the reviewer that lumbar puncture with CSF analyses including neurofilament light chain levels, alfa synuclein etc, is important for the differential diagnosis. However, the patient refused lumbar puncture due to her fear in worsening of her chronic lumbar muscle strain after the examination. As a matter of fact, the patient’s symptoms, MRI imagings and biopsy results were compatible with NIID. We could exclude major CNS infectious diseases and autoimmune encephalitis, as the patient did not report any symptoms such as fever, headache, mental and behavior disorder.

Point 2: page 3 ln 110-111: “Her cognitive function was intact except for some short-term memory deficits”. How short-term memory was assessed? (eg. Tower of London). If there are no objective evidence of short-term memory deficits , this sentence should be removed. Short-term memory does not refers to a slight forgetfulness of recent events.

Response 2: We thank the reviwer for advice. We have now removed this sentence “except for some short-term memory deficits” (page 3, line 111).

Point 3: page 3 ln 125-127: “There were 125 no obvious abnormalities in the following: routine blood tests, biochemical examination 126 of blood plasma, infectious indicators, tumour markers, thyroid hormone test or immunological index.” You should report what exams were performed (eg. immunological index ? what index? ANA, ANCA?)

Response 3: Thank you for your professional comments. The results of laboratory tests including complete blood count, urine and stool analyses, liver and kidney function, lipid profile, and blood glucose concentration were unremarkable. Infectious indicators (including HIV, treponema pallidum, hepatitis virus, blood culture, purified protein derivative test, galactomannan test, and glucan test), hormone test or immunological index (including ANA, ANCA, IgA, IgG, IgM, IgE, complement-3/4, λ and κ light chain immunoglobulin), tumour markers (including carbohydrate antigen 724 (CA724), carcinoembryonic antigen (CEA), alphafetoprotein (AFP), neuron-specific enolase (NSE), CA19-9, CA125, CA153, β-human chorionic gonadotrophin (hCG), cytokeratin-19-fragment (CYFRA21-1), CA50, human epididymis protein 4 (HE4), risk of ovarian malignancy algorithm (ROMA), squamous cell carcinoma antigen (SCCAg) ), and all were proved negative. These descriptions have been added to the revised manuscript and highlighted in red (page 3-4, lines 126-136).

Point 4: MRI: other sequences may be reported such as FLAIR, ADC, T2, T1 gd+. Indeed, DWI indicates the presence of recent ischemic stroke (cytotoxic edema) or demyelinating active lesions. Reporting the other MRI sequences would help the readership to better understand the nature of the disease.

Response 4: We thank the reviwer for the comments. We have now added description of T1-weighted (page 4, lines 130-140) and ADC imaging (page 4, lines 143-144), and highlighted in red. Axial planes of T2-weighted and fluid attenuated inversion recovery (FLAIR) images of the brain using magnetic resonance imaging (MRI) revealed symmetrically diffuse, high signal leukodystrophy. The area presented isointense or slightly hypointensities in the T1-weighted imagings. Diffusion-weighted-imaging (DWI) sequence showed a high signal intensity zigzag edging sign in the bilateral fronto-parieto-occipital corticomedullary junction. The corresponding lesions showed isointense or slightly hyperintense signals on ADC sequence. 

Point 5: page 5 ln 172-183: “We reviewed the pathogenic factors of the secondary neu- 173 rotrophic along the pathways of keratitis corneal sensory nerve, including the following 174 causes: (1) injuries of corneal sensory nerves induced by viral infection, refractive surgery, chemical or thermal burns, ocular medication and wearing contact lenses; (2) damage of 176 the ophthalmic branch of the trigeminal nerve as well as ciliary nerves due to tumour 177 formation and surgeries (such as cataract and retinal surgeries or ophthalmic laser procedures); (3) trigeminal ganglion lesions due to herpetic viral infections, surgery for trigeminal neuralgia, preganglionic trigeminal nerve root injury (including acoustic neurinoma, 180 schwannoma and aneurysms) and some systemic diseases (such as diabetes and leprosy) 181 that can decrease sensory nerve function or damage sensory fibres and) damage to the lowest part of spinal trigeminal nuclei due dorsolateral medullary infarction and tumour formation”. The diagnostic process should be reported in the section of “2.Case Report (eg. results of viral serology, HSV, VZV) thus this paragraph may be shortened.

Response 5: We thank the reviwer for this constructive comment. We agree with you and have revised to make it more reasonable along the text and highlight with red. We have modified and integreted this part to the section of diagnostic process (page 4, lines 158-171), and have simplified it in the Discussion section (page 5, lines 198-211). The order of the cited references was adjusted. 

Point 6: page5 ln 220-221 “For example, the premutation of the FMR1 gene leads to neuro- 220 degeneration and the development of intranuclear inclusions in NIID [22, 24]. “ Ref n 24 is lacking in the manuscript. However, this sentence may be deleted since you have already said it elsewhere in the manuscript.

Response 6: We agree with the reviwer and have deleted this sentence. 

Point 7: language revision /grammar check

Response 7: The original manuscript had gone through language check service before submission. The language and grammar have been re-checked now.

Reviewer 2 Report

I read the manuscript with great interest.

How about the genetic analysis of NOTCH2NLC?

Authors excluded FXTAS by the genetic analysis of FMR1.

The genetic analysis of NOTCH2NLC will add even more significance to the report.

Author Response

We gratefully acknowledge the comments raised by the reviewer. These remarks have helped to improve the manuscript significantly. The comments have been carefully considered and addressed below.

Point 1: How about the genetic analysis of NOTCH2NLC? The genetic analysis of NOTCH2NLC will add even more significance to the report. 

Response 1: We agree with the editor that a genetic analysis may assist diagnosis. Indeed, GGC repeats of the NOTCH2NLC were found to be associated with sporadic NIID (Tian et al., 2019; Ishiura et al., 2019). Howeve, in this case, the patient and her family refused the whole-genome sequencing due to the high expense. Nevertheless, the characteristic symptoms, brain MRI manifestation and skin biopsy enable a diagnosis of definite NIID.